# Artificial Intelligence in Coronary Artery Calcium Scoring Detection and Quantification

**DOI:** 10.3390/diagnostics14020125

**Published:** 2024-01-05

**Authors:** Khaled Abdelrahman, Arthur Shiyovich, Daniel M. Huck, Adam N. Berman, Brittany Weber, Sumit Gupta, Rhanderson Cardoso, Ron Blankstein

**Affiliations:** Departments of Medicine (Cardiovascular Division) and Radiology, Brigham and Women’s Hospital, Harvard Medical School, Boston, MA 02115, USA

**Keywords:** coronary artery calcium, artificial intelligence, atherosclerosis, computed tomography

## Abstract

Coronary artery calcium (CAC) is a marker of coronary atherosclerosis, and the presence and severity of CAC have been shown to be powerful predictors of future cardiovascular events. Due to its value in risk discrimination and reclassification beyond traditional risk factors, CAC has been supported by recent guidelines, particularly for the purposes of informing shared decision-making regarding the use of preventive therapies. In addition to dedicated ECG-gated CAC scans, the presence and severity of CAC can also be accurately estimated on non-contrast chest computed tomography scans performed for other clinical indications. However, the presence of such “incidental” CAC is rarely reported. Advances in artificial intelligence have now enabled automatic CAC scoring for both cardiac and non-cardiac CT scans. Various AI approaches, from rule-based models to machine learning algorithms and deep learning, have been applied to automate CAC scoring. Convolutional neural networks, a deep learning technique, have had the most successful approach, with high agreement with manual scoring demonstrated in multiple studies. Such automated CAC measurements may enable wider and more accurate detection of CAC from non-gated CT studies, thus improving the efficiency of healthcare systems to identify and treat previously undiagnosed coronary artery disease.

## 1. Introduction

### 1.1. Artificial Intelligence in Medical Imaging

Artificial intelligence (AI) tools have experienced accelerated growth in recent years across various areas of radiology, including cardiac imaging [1,2]. Further, the use of AI in medical imaging has generated enormous excitement surrounding its potential to transform clinical practice by assisting in image acquisition and interpretation, as well as improving image quality and the detection of disease [3]. Additionally, AI applications have been shown to improve risk prediction and stratification across various disease states and disciplines with wide-ranging applications [4]. The use of artificial intelligence applied to cardiovascular imaging via “radiomics” may further enhance detection of disease and ultimately improve prognosis [5,6]. AI has been implemented across all imaging modalities and is being used in image segmentation and automated measurements, including the automated assessment of coronary artery calcium (CAC) [7,8,9].

These developments have been facilitated by improvements in imaging technology and AI techniques, including deep learning [8]. AI refers to systems that perform tasks that typically require human intelligence and encompass the fields of machine learning and deep learning. Machine learning is a branch of AI that includes dynamic systems that learn from data and algorithms without explicit programming [10]. Deep learning is a branch of machine learning that utilizes computational models inspired by biological neural networks and is increasingly used in contemporary AI approaches [11].

### 1.2. Coronary Artery Calcium Assessment and Interpretation of Non-Cardiac Exams

Coronary artery calcium (CAC) assessment is a robust predictor of future atherosclerotic cardiovascular disease (ASCVD) risk and is currently endorsed by numerous cardiovascular guidelines to enhance risk assessment [12,13,14]. The prognostic value of CAC, which is a marker of prevalent atherosclerosis, has been demonstrated in several diverse, large-scale prospective studies, including the Multi-Ethnic Study of Atherosclerosis (MESA), Framingham Heart Study (FHS), Dallas Heart Study, and the CAC consortium [15,16,17]. CAC can help guide the potential benefit of statin therapy and is increasingly used in shared decision-making, especially among asymptomatic intermediate-risk individuals without known ASCVD [18]. Multiple clinical practice guidelines have endorsed the use of CAC for targeted risk stratification in selected patients, including the AHA/ACC, ESC/EAS, and the National Lipid Association (NLA). The use of CAC for risk stratification has direct implications for patient care, as the identification of coronary atherosclerosis has been shown to result in the initiation or intensification of lipid-lowering and other preventive therapies [17,19,20]. Recent studies have also used CAC as a selection criteria for novel cardiometabolic agents, lipid-lowering therapies, and anti-inflammatory agents as part of trial entry criteria as a potential marker for patients at high risk of ASCVD events.

In addition to dedicated CAC scans, which use ECG gating, CAC can be detected on traditional non-gated non-contrast chest CT (NCCT) examinations that are performed for other clinical indications. These include low-dose chest CT scans performed for lung cancer screenings, which are performed approximately 7.1 million times annually in the US [21,22]. Importantly, CAC scoring from these non-ECG-gated exams correlates well with dedicated studies and similarly predicts adverse cardiovascular outcomes [23,24,25,26,27,28]. The Society of Cardiovascular Computed Tomography (SCCT) and the Society of Thoracic Radiology (STR) guidelines recommend the reporting of CAC on all NCCT examinations, although such evaluation and reporting of CAC on non-gated CT chest exams is only rarely performed [29,30]. 

A significant proportion of NCCT studies are performed for lung cancer screening [31]. Because of their age and smoking history, this population of patients is nearly all at intermediate to high risk for coronary artery disease (CAD) [32]. However, CAC reporting from NCCT is variable. In one study, CAC was present on NCCT in 58% of patients, but only described in the radiology report of 44% of these cases [33]. In another study, 53% of patients were found to have CAC, but CAC was only reported in 59% of these cases [34]. Recognition of CAC on NCCT is important because identification of CAC has been associated with interventions that may enhance the prevention of CVD [35]. Thus, automation of CAC scoring in NCCT holds numerous advantages (Figure 1) and may serve to increase the recognition, reporting, and accuracy of CAC detection on non-ECG gated scans. 

### 1.3. Artificial Intelligence and CACS

CAC scoring, as originally described by Agatson and colleagues in 1990, is a manual process [36]. By this original method, CT images are reviewed in axial slices, and areas of calcification are manually selected by the reader. These lesions are then quantified by a product of area x density factor, according to the Hounsfield unit. The total calcium score is the sum of the scores of individual lesions [36]. Current methods of CAC analysis are semi-automated and require post-processing after image acquisition, including specialized software [37,38]. This process requires time and additional work and thus poses challenges for large-scale evaluation of CAC.

The automation of CAC scoring by artificial intelligence methods is highly desirable. The use of artificial intelligence-assisted CAC scoring is not affected by human factors and improves intraobserver and interobserver variability. Additionally, AI models are much faster and could offer improved detection compared with human readers [37]. 

## 2. AI Applications for CAC Scoring from Cardiac/Gated Scans

### 2.1. Early ML-Based Approaches

A number of methodologies, including rule-based models, machine learning, and deep learning, have been developed to automate CAC scoring from ECG-gated cardiac scans [39]. Early CAC automation techniques used supervised machine learning approaches, including k-nearest neighbor (KNN) classifiers and support vector machines (SVM) [40]. These early pattern recognition approaches focused on first selecting candidate coronary calcific lesions through feature-based extraction, followed by various classifier-based approaches to distinguish true coronary calcifications from other bystander calcifications in the thorax.

One early attempt to automate coronary calcium scoring in ECG-gated studies was described by Išgum et al. in 2007. In their approach, candidate coronary calcifications were extracted via thresholding and component labeling, with features extracted including size, shape, and spatial position relative to the heart and aorta, and without extraction of the coronary arteries. Classification systems using these features were used to determine which of these objects represented coronary calcifications, with a calcium score and risk category ultimately reported. This early approach resulted in a sensitivity of approximately 74% for detecting coronary calcifications, with a false-positive rate of 10% per scan on average [41]. Kurkure et al. also utilized a classification-based approach to detect coronary calcifications from cardiac CT scans. Their classifiers first distinguished true vascular calcification (coronary and aortic) from other high-density objects and, in a second stage, separated aortic calcifications from coronary calcifications. Across 105 subjects, their methods demonstrated high sensitivity (92.1%) and specificity (98.6%) [42]. In a subsequent study, an automated system was developed that identified the specific coronary artery with which calcifications were associated. This approach included feature-based extraction followed by a supervised hierarchical classification approach to identify coronary artery calcifications. Coronary artery locations were estimated using an atlas-based method built from 85 CT angiography scans. The sensitivity of calcifications in this study was 86.6% in a 3.0 mm data set and 81.2% in a 1.5 mm dataset [43]. Another study using a rule-based approach by Ding et al. was reported in 2015 and presented an automated algorithm for calculating calcium scores. This demonstrated a high correlation with manual Agatson scores (R = 0.97, *p* < 0.001), with a total computing time of less than 60 s [44]. While these initial approaches demonstrated promise in the early automation of CAC scoring, advances using deep learning and other AI-related technologies have improved CAC automation.

### 2.2. Deep Learning and Convolutional Neural Networks 

Deep learning approaches have been utilized in automated CAC analysis in dedicated ECG-gated cardiac CT. Most deep learning approaches are based on artificial neural networks (ANN), so-called because of the resemblance of their design to biological neurons. The most common and successful deep learning technique for extracting features from raw medical imaging data is convolutional neural networks (CNN), which has been applied to CAC automation and is the state-of-the-art approach in detection and segmentation in image processing [45]. 

A critical contribution to the rapid development of deep learning and computer vision was the development of AlexNet, a large CNN designed by Krizhevsky et al. to classify 1.2 million high-resolution images with very high performance [46,47,48]. As compared with machine learning approaches previously described, CNN methods classify individual voxels rather than individual candidate calcific lesions [40]. 

Various statistical measures of interrate measurement have been utilized to evaluate the accuracy of different models. These include kappa, utilized for ordinal outcomes (e.g., risk categorization), and intraclass correlation coefficient (ICC) for continuous measures (e.g., calcium score) [49]. The kappa statistic adjusts the observed agreement between two observers classifying subjects into ordinal categories, subtracting and normalizing the agreement attributed to chance alone [50]. With respect to CAC, this refers to risk categorization based on calcium score (e.g., CAC score 0, 1–10, 11–100, 101–400, >400), which has implications in clinical management [51]. ICC, on the other hand, is a widely used reliability measure that assesses the similarity of repeated measurements within a class of data between two different raters [52,53].

In one study, a CNN deep learning model to measure CAC on 79 scans showed near-perfect agreement with manual scoring (difference in scores = −2.86, Kappa = 0.89, *p* < 0.0001) and improved speed compared with the manual method [38]. Larger deep learning models based on CNN have applied these techniques to larger cohorts, such as U-Net, developed by Hong et al., which detected CAC automatically in 1811 cases with a sensitivity of 99%, specificity of 100%, and intraclass correlation coefficient (ICC) of 1.00 between standard and model-predicted scores. The speed of CAC detection was only 50 ms per CT [37]. Sandstedt et al. included 315 CAC scans in their study, which compared semi-automatic and automatic software for CAC score assessment. They demonstrated strong agreement between the Agatson score (rho = 0.935), volume score (rho = 0.932), and mass score (rho = 0.934) between the methods [54].

In a larger study, Martin et al. used CNN in combination with residual networks (ResNet) using a research prototype called Automated CAScoring (Siemens Healthineers) to evaluate 511 CAC scans [55,56]. Their results had excellent agreement with the reference standard (Spearman rho = 0.97 and ICC 0.985), with agreement on risk categorization in 93.2% of patients between human and automated classification [55]. When applied to a larger dataset of 1171 CAC studies from a multicenter dataset, the Siemens Healthineers algorithm demonstrated 97% sensitivity, 93% specificity, and 94% accuracy in branch label specification. This study demonstrated a median absolute Agatston score difference of 0 (IQR 0.0–1.3) [57]. Further demonstrating the feasibility of fully automated CAC scoring using DL in a larger study, Idhayid et al. tested their multiple-CNN algorithm in 1849 scans. Their model demonstrated increased detection of CAC (47%) compared to human readers, although 9% of individuals with 0 CAC were reported as having a positive score by AI [39]. This was attributed to the detection of non-coronary calcifications and noise. Additional studies utilizing deep learning techniques for automating CAC scoring from gated scans are described in Table 1.

## 3. AI for CAC Analysis on Non-Gated CT Scans 

Given the prognostic value and clinical impact of calcium scoring, there is increased interest in reporting coronary calcium scores from non-gated chest CT studies performed for other reasons [57]. Given the large number of chest scans performed, there is a significant opportunity to improve cardiovascular risk assessment for patients who have had scans performed previously for other reasons. An example of calcium detected on a cardiac and non-gated chest CT is shown in Figure 2. Additionally, joint guidelines from major societies advocate that CAC should be reported on all non-contrast CT examinations of the chest [29]. Artificial intelligence algorithms—and particularly deep learning for CAC automation—have been developed for non-gated scans and compared with dedicated ECG-gated CAC scans, and offer significant advantages (Box 1).

Box 1AI-based incidental detection of coronary artery calcium from non-cardiac CT has numerous advantages, from detection to risk assessment and improving population-based prevention efforts.Box: Advantages of Al based incidental detection of CAC from non-cardiac CT
 •Improve ability to detect presence and burden of CAC •Improve reproducibility and accuracy of CAD detection •Enhance risk assessment, thereby guiding need & intensity of preventive therapies •Improve population-based preventive efforts

One early demonstration of this application used a machine-learning k-nearest neighbor approach to analyze 1749 non-contrast non-gated low-dose chest CT scans, with a reliability of k = 0.85 with respect to manual Agatson risk categorization and a mean difference of 2.5 for the Agatston score (ICC = 0.90) [60]. Automation in non-gated scans has also been applied using deep learning techniques. In 2016, Lessmann et al. trained three independent CNN models using 797 non-contrast, non-ECG-gated chest CT scans. The models were then applied to 231 scans of the same type, with 97.2% of coronary calcifications detected and 84.4% accuracy in risk category assignment (kappa = 0.89) [61]. 

One larger example utilized pulmonary non-ECG-gated CT from the COPDGene study. Deep CNN was applied to 5973 images from this dataset, with 1000 of those comprising the test set and 4973 comprising the training set. The algorithm achieved a high Pearson correlation for computed scores compared to the reference standard (rho = 0.932, *p* < 0.0001), with 75.6% of patients assigned to the correct risk stratification [62]. Van Assen et al. used a CNN with a ResNet architecture for the image features, along with a separate fully connected neural network for spatial features. This algorithm (AI-Rad Companion Chest CT from Siemens Healthineers, Erlangen, Germany) was trained on 95 gated studies and then refined on non-gated CT chest studies. Subsequently, the algorithm was tested on 168 patients who underwent chest CT examinations, with a high correlation between the manual Agatston score and calcium volume (Pearson correlation coefficient = 0.921, *p* < 0.001) and 91% sensitivity and 92% specificity to detect calcium; 82% of cases were classified in the correct risk category (kappa = 0.74) [63].

One challenge is demonstrating generalizability to deep-learning models across scanner types and hospital systems. One example of working to improve the transportability and generalizability of this technology includes a deep-learning software (CACScoreDoc, Shukun Technology, Beijing, China), which demonstrated the ability to calculate CACS based on 901 chest CT scans from multiple scanner vendors and multiple hospitals [64]. Additionally, all patients had also undergone ECG-gated CT scans, and CAC scoring was compared using the manual method applied to gated scans and the automated method applied to routine chest CT studies, with a strong correlation between AI-assisted and manual CAC (rho = 0.893, *p* < 0.001) as well as risk category agreement (kappa = 0.679, *p* < 0.001, concordance of 80.6%). This study further demonstrated the applicability of such an algorithm across various scanners and protocols, an important feature for its implementation into routine clinical practice.

In a 2020 study by van Velsen et al., 7240 scans from various types of examinations were collected, including chest CT, PET attenuation correction CT, radiation therapy planning CT, CAC screening CT, lung screening CT, and other low-dose CT of the chest, demonstrating a wide range of protocols and subject variability. This study used CNN-based deep learning to evaluate calcium scores and showed sufficient agreement with manual scoring (ICC 0.79–0.97) with improved ICC (ICC 0.84–0.99) when additional protocol-specific CT scans were added [65]. Importantly, this multicenter study demonstrated a robust algorithm that performed well across various types of scans, scanners, and protocols. Zelzenik et al. collected a mix of gated and non-gated scans across a variety of protocols and scanners from multiple datasets, including the Framingham Heart Study, the National Lung Cancer Screening Trial, PROMISE, and ROMICAT-II. This deep learning algorithm demonstrated very high agreement between automated and manual scores (Spearman rho = 0.92, *p* < 0.0001) and concordance in risk stratification between automated and manual methods (Kappa = 0.70, concordance rate 0.79) [66]. Taken together, these studies highlight the significant potential to apply automated CAC scoring to non-gated routine CT scans conducted for a wide range of clinical indications and demonstrate a potential role for opportunistic screening.

## 4. Additional Applications

There is also potential for AI to be applied to other types of scans than traditional chest CT scans to quantify CAC. For example, chest radiographs have been used to predict coronary artery calcifications using deep-learning CNN methods. In one study, 1689 chest X-rays were assessed in patients who also underwent cardiac CT in the same year to assess for the presence of coronary calcium [67]. A binary classification of zero vs. nonzero total Agatson score was achieved with 91% sensitivity but only 29% specificity on frontal chest x-rays when optimized for sensitivity. However, given these test characteristics, this technology would be unlikely to serve as a strong screening tool prior to CAC in its current state. Fully automated CAC scoring has also been demonstrated from non-contrast ungated scans from 18F-FDG-PET/CT and does not require changes to the PET/CT scanning protocol [68,69].

CAC scores can also be predicted indirectly using imaging that does not encompass the coronary arteries. For example, a model for the prediction of coronary atherosclerosis has been applied to retinal photographs using deep learning. A deep-learning algorithm called RetiCac improved the prediction of atherosclerosis with an AUC of 0.742 [70]. Additionally, calcium scoring has been automated and applied to patients undergoing myocardial perfusion imaging, with a high correlation of calcium scoring between automatic and manual techniques (weighted kappa 0.95, 95% CI 0.92–0.97), which was applied to 213 patients, although this technique had a low negative predictive value [71]. Mu et al. trained their deep learning method to automate CAC scoring from 240 coronary CTA scans and demonstrated a very high correlation between CTA and non-contrast CT scans (Pearson = 0.96) and a risk categorization agreement of kappa = 0.94 [72]. Wolterink et al. applied paired CNN to identify calcified voxels and automate CAC scoring from coronary CTA in 2016 in a study of 250 patients in which coronary CTA and CAC had been performed. Their study detected 72% of lesions in a test set and demonstrated that CAC can be automatically quantified from CCTA [73].

## 5. Future Directions for AI in CAC Scoring

The detection and quantification of CAC scores from non-gated cardiac CT shows great potential for improving ASCVD risk estimation and guiding the potential need for preventive therapies (Figure 3). In particular, deploying accurate deep learning systems to large imaging databases can enhance the efficiency of healthcare systems to identify previously undiagnosed coronary artery disease. In order to gain wide acceptance, such techniques need to be feasible to implement and automated so that they do not add additional interpretation time. Importantly, the identification of plaque can either be performed prospectively (i.e., at the time that images are interpreted by a radiologist) or retrospectively (i.e., from previously acquired images). The adoption of AI-based CAC detection in a prospective manner will require the integration of automated CAC scoring with the radiologist’s workflow, which may increase the recognition and reporting of CAC as well as improve the accuracy of reporting CAC severity. Ultimately, if AI-based CAC detection is implemented across different settings, the ability to detect plaque from any CT could improve the automated prediction of cardiovascular risk and serve as a large-scale use of AI to improve cardiovascular health. 

Automated CAC measurement has also been applied to assess cardiovascular risk and the risk of all-cause mortality. For example, a DL CAC algorithm was applied to 428 patients undergoing non-contrast CT with locally advanced lung cancer. Increased CAC was associated with increased all-cause mortality in these patients, suggesting the potential predictive application for this patient population [74]. One example of this is from Zeleznik et al. Their group applied their previously described DL algorithm to automatically quantify CAC from 20,094 asymptomatic individuals from the Framingham Heart Study and National Lung Screening Trial (NLST), as well as individuals with chest pain syndromes from the PROMISE and ROMICAT-II studies. As mentioned previously, their algorithm was applied to a wide range of CT scan protocols and scanners, including ECG-gated and non-gated scans, with 5521 subjects assessed by human readers for reference. In patients from NLST in the Zeleznik study, the automated CAC algorithm was applied to 14,959 patients. At the median 6.7-year follow-up, those with high-risk coronary calcium had an increased hazard of ASCVD-related outcomes compared to the reference group. For example, the high-risk calcium group had a HR of 3.87 (95% CI = 2.45–6.11, *p* < 0.001) compared with the reference group [66].

Further, a recent study of 5678 adults without ASCVD used a DL algorithm on a non-gated chest CT to calculate CAC. They determined that incidental CAC > 100 detected by the DL algorithm was associated with increased all-cause mortality and adverse CV outcomes, including in adjusted analyses. Additionally, they found that only 26% of those with CAC > 100 detected by the algorithm were on statin therapy, demonstrating the potential of the tool for facilitating earlier intervention [75].

When applied to stable chest pain patients from the PROMISE trial, deep learning calcium score was strongly associated with cardiovascular events, with increases in hazard across deep learning calcium score categories. This large study, applied to data from major trials, demonstrated the significant potential of deep learning to help refine ASCVD risk from gated or non-gated studies at a disease- and population-specific level. Larger-scale studies with higher-performing algorithms are needed for a broader implementation of automated CAC to be adopted.

Numerous strategies to improve coronary calcium analysis have been proposed, including inverse weighting of calcium density, assessment of the number or size of calcified plaques, or the regional distribution of calcifications [76,77,78,79]. Additional strategies could include the incorporation of extracoronary features, such as extra coronary calcifications, which are associated with ASCVD events, or the presence of epicardial fat or hepatic steatosis, which are associated with increased severity of coronary calcifications [80,81,82,83]. Such approaches could improve the correlation of CAC scoring with total coronary plaque burden or high-risk plaque characteristics, but they are time-consuming and computationally challenging at this time [79]. Ultimately, the use of artificial intelligence may provide additional features that, together with CAC, will offer robust and clinically actionable data for guiding preventive therapies.

## 6. Potential Challenges to Automated CAC Scoring

There remain several challenges to the automation of CAC scoring using artificial intelligence. First, automation of CAC must be performed using a workflow that will be efficient and will not require extra work by radiologists. Second, potential AI solutions need to perform accurately across different types of scanners and imaging protocols. Third, AI algorithms need to be able to distinguish between non-coronary calcifications such as valvular calcification and other high-density objects (e.g., metal implants) from coronary calcifications. Other challenges include the exclusion of various other sources of noise and artifacts from tissues surrounding the coronary arteries [8,41].

## Figures and Tables

**Figure 1 diagnostics-14-00125-f001:**
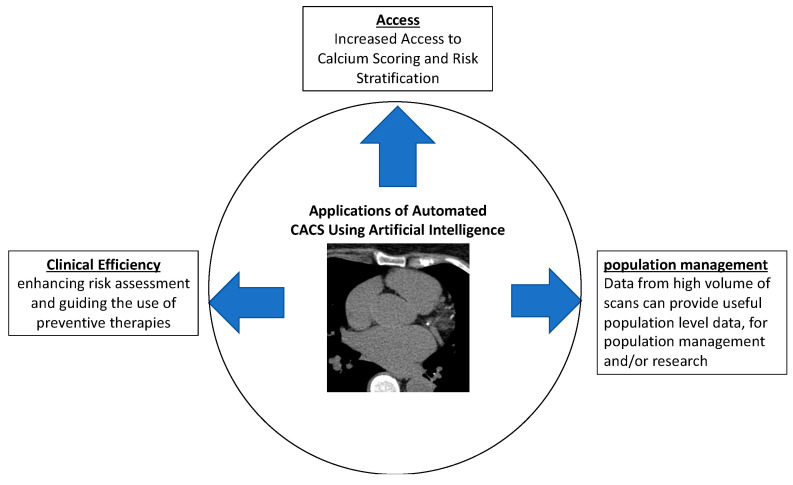
There are widespread applications of automated calcium scoring using artificial intelligence approaches. These include improved clinical efficiency to guide risk assessment and preventive therapy, increased access to calcium scoring, and increased population-level data from a high volume of scans to which the technology can be applied.

**Figure 2 diagnostics-14-00125-f002:**
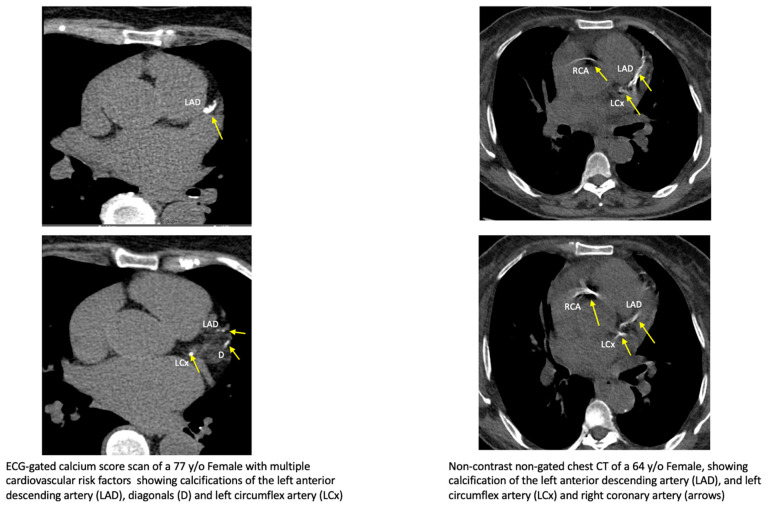
Example of a coronary calcium cardiac scan (**left**) is demonstrated alongside a non-contrast chest CT from a patient demonstrating coronary calcifications (**right**).

**Figure 3 diagnostics-14-00125-f003:**
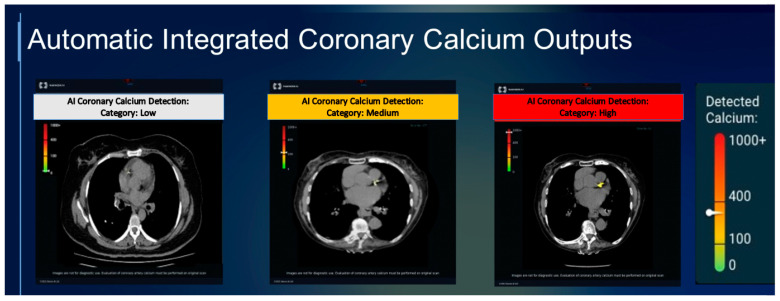
Examples of AI-based detection of CAC on non-contrast chest CT. The AI algorithm shown quantifies the coronary artery calcium (CAC) score and categorizes the amount of plaque as low, medium, or high. Figure courtesy of Nanox AI Ltd. (Petah Tikva, Israel).

**Table 1 diagnostics-14-00125-t001:** Deep learning techniques for automating CAC from gated scans.

Study	Year Published	Study Size (Testing)	Algorithm Type	Risk Categories	Accuracy	Conclusions
Eng et al. [38]	2021	79	CNN	Categories for Agatston scores of 0, 1–10, 11–100, 101–400, >400	Mean difference scores: −2.86, Kappa = 0.89, *p* < 0.0001	Demonstrated near-perfect agreement with the reference standard and with improved computational speed.
Hong et al. [37]	2022	959	U-Net (CNN)	Categories for Agatston scores of 0, 1–10, 11–100, 101–400, >400	ICC = 1.00, Kappa = 0.931	Demonstrated excellent agreement with the reference standard and detected mild calcifications not detected by reference.
Gogin et al. [45]	2021	98	U-Net (CNN)	Categories for Agatston scores of 0, 1–10, 11–100, 101–400, >400	Concordance-index = 0.951	With an ensemble of 5 CNN models, there is high concordance with the standard reference.
Zhang et al. [58]	2021	46	U-Net (CNN)	Risk categorization not compared	ICC = 0.988, mean difference scores: −6.7, *p* = 0.993	High-speed and accurate automated quantification of total and vessel-specific CAC in a single-center study.
Sandstedt et al. [54]	2020	315	CNN	Categories for Agatston scores of 0, 1–10, 11–100, 101–400, >400	Mean difference scores: −8.2, ICC = 0.996, Kappa = 0.919	Single-center study demonstrating near-perfect agreement, including Agatson assessment, volume score, mass score, and number of calcified lesions.
Wang et al. [59]	2019	140	3D CNN	Categories for Agatson scores of 0, 1–99, 100–299, and >300	ICC = 0.94, Kappa = 0.77	Single-center study with near-perfect agreement of Agatson, volume, and mass scores and a reclassification rate of 13%.
Martin et al. [55]	2020	511	ResNet CNN	Categories for Agatston scores of 0, 1–10, 11–100, 101–400, >400	ICC = 0.985, Kappa = 0.932	Demonstrated outstanding agreement of total Agatson score with the reference standard trained on a dataset of 2000 patients.
Winkel et al. [57]	2022	1171	CNN	Categories for Agatston scores of 0, 1–10, 11–100, 101–400, >400	ICC = 0.84, Kappa = 0.91	Large, multicenter study demonstrating excellent accuracy on a total and per-vessel basis.
Idhayid et al. [39]	2022	1849	3D CNN	Categories for Agatston scores of 0, 1–10, 11–100, 101–400, >400	ICC = 0.98, Kappa = 0.90, *p* < 0.001	Large study with scans obtained from multiple vendors demonstrated excellent agreement and efficiency.

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
