# Peer review of "Artificial Intelligence in Coronary Artery Calcium Scoring Detection and Quantification"

_diagnostics, 2024, doi:10.3390/diagnostics14020125_

Round 1

Reviewer 1 Report

Comments and Suggestions for Authors

See the attached file for the suggested revisions

I recommend adding a section about the potential challenges of the technology that can be merged with the future perspectives.

Some of the potential challenges are already described in the text, but it should be better a distinct section for this topic

In the conclusions further underline the potential of the integration of an automated CAC in the radiologists workflow for the potential patients'benefits and a more personalized therapeutic approach

Comments on the Quality of English Language

See the attached file for the suggested revisions

Reviewer 2 Report

Comments and Suggestions for Authors

The paper "Applications of Artificial Intelligence in Coronary Artery Calcium Scoring Detection and Quantification" discusses the use of AI in detecting and quantifying coronary artery calcium (CAC), a marker of coronary atherosclerosis. It highlights the importance of CAC as a predictor of cardiovascular events and how AI, including machine learning and deep learning approaches, has improved the efficiency and accuracy of CAC scoring. The paper reviews various AI techniques used for this purpose, their development over time, and their implementation in different imaging modalities. It also explores the potential of AI in enhancing healthcare efficiency by enabling more accurate detection and treatment of coronary artery disease. The paper is comprehensive and well-structured, providing detailed insights into the current state and future potential of AI in cardiac imaging.
